

# Hematopoietic stem cell discovery: unveiling the historical and future perspective of colony-forming units assay

Nur Afizah Yusoff, Zariyantey Abd Hamid, Siti Balkis Budin and Izatus Shima Taib

Biomedical Science Programme and Center for Diagnostic, Therapeutic and Investigative Studies, Faculty of Health Sciences, Universiti Kebangsaan Malaysia, Kuala Lumpur, Malaysia

## ABSTRACT

Stem cells are special cells with the distinctive capability to self-renew, forming a new pool of undifferentiated stem cells. They are also able to differentiate into lineage-specific cell types that are specialized and matured. Thus, stem cells are considered as the building blocks of tissues and organs in which they reside. Among the many types of stem cells, hematopoietic stem cells (HSCs) are the most studied adult stem cells and are considered as a promising source of cells for applications in the clinical and basic sciences. Historically, research on HSCs was initiated in the 1940s, where in a groundbreaking experiment, intravenously injected bone marrow (BM) cells prevented the death of irradiated mice by restoring blood cell production. Since then, HSCs have been studied and utilized in medical therapies and research for over several decades. Over time, more sophisticated tools have been developed to evaluate the behaviour of specifically purified subsets of hematopoietic cells that have the capacity to produce blood cells. One of the established tools is the colony-forming units (CFUs) assay. This assay facilitates the identification, enumeration, and analysis of colonies formed by differentiated hematopoietic stem and progenitor cells (HSPCs) from myeloid, erythroid and lymphoid lineages. Hence, the CFUs assay is a fundamental *in vitro* platform that allows functional studies on the lineage potential of an individual HSPCs. The outcomes of such studies are crucial in providing critical insights into hematopoiesis. In this review, we explore the fundamental discoveries concerning the CFUs assay by covering the following aspects: (i) the historical overview of the CFUs assay for the study of clonal hematopoiesis involving multilineage potential of HSPCs, (ii) its use in various experimental models comprising humans, mice/rodents, zebrafish and induced pluripotent stem cells (iPSCs) and (iii) research gaps and future direction concerning the role of CFUs assay in clinical and basic sciences. Overall, the CFUs assay confers a transformative platform for a better understanding of HSPCs biology in governing hematopoiesis.

Corresponding author
Zariyantey Abd Hamid,
zyantey@ukm.edu.my

## INTRODUCTION

The understanding of hematopoiesis and hematopoietic stem cells (HSCs) has evolved significantly over the past two centuries. The late 1860s marked the beginning of the formal scientific description of hematopoiesis. Before this period, the understanding of blood and its formation was rudimentary, with earlier work focusing more on the general composition of blood and circulation rather than the specific processes of hematopoiesis. Neumann and Bizzozero were credited with the discovery that bone marrow (BM) is the primary site of blood cell formation. In 1868, both scientists published findings that linked BM to hematopoiesis, revolutionizing the understanding of blood formation at that time (*Neumann, 1868*; *Bizzozero, 1868*). Ehrlich, a German scientist, further advanced the field from 1880s by introducing new staining techniques that allowed for the differentiation of various blood cells. His work helped classify different types of white blood cells, which was crucial for the understanding of hematopoiesis (*Ehrlich, 1888*). The modern concept of HSCs was established through the pioneering work of *Till & McCulloch (1961)*. Their experiments demonstrated the existence of stem cells capable of regenerating the entire blood system. They used radiation experiments in mice to show that single cells in the BM could proliferate and differentiate into various blood cell lineages, leading to the formation of colonies in the spleen (*Till & McCulloch, 1961*). This work laid the foundation for the modern understanding of HSCs and their central role in hematopoiesis.

It is no understatement that the discovery of HSCs have transformed and reshaped conventional thinking about the origins of hematopoiesis. Moreover, HSCs confer great potential as powerful healing tools and building blocks in the clinical sciences. The genesis of HSCs niches, initially within the hematopoietic system of the developing embryo and subsequently in the BM throughout life, has captivated scientists for centuries as they investigated the biological properties of HSCs. While HSCs are rare and long-lived, there are many intermediate progenitor cell types that in a stepwise manner, lose their multilineage and self-renewal potency before becoming mature functioning blood cells (*Kauts, Vink & Dzierzak, 2016*). Two main branches of the hematopoietic lineage namely the common myeloid progenitors (CMPs) and common lymphoid progenitors (CLPs) are derived from multipotential progenitors (*Weiskopf et al., 2016*). The ideal microenvironment for HSCs make them able to survive, regenerate themselves through their self-renewal property and differentiation into specific lineage-committed progenitors that undergo subsequent terminal differentiation into various mature blood cells including B and T lymphoid cells, dendritic cells, natural killer cells (NK), granulocytes, monocyte/macrophages, red blood cells and platelets (*Cheng, Zheng & Cheng, 2020*). Thus, hematopoietic stem and progenitor cells (HSPCs) are considered as the foundation of the body's blood cell production system, playing a pivotal role in maintaining overall health and immune function (*Zhu et al., 2020*).

Historically, HSCs research as a quantitative science emerged as a by-product of other investigative strategies seeking to determine how the complication of myeloablation on patients can be prevented or minimized. This functional approach to identify HSCs began with the discovery that their regenerative capacity can be triggered when transplanted into a host whose blood cell production has been impaired, such as by exposure to a normally

lethal dose of radiation. *Jacobson et al. (1951)* suggested that recovery from radiation is likely driven by a humoral factor rather than by migrating cells or detoxification by shielded or transplanted tissues. This points to the involvement of a substance rather than cellular mechanisms in the recovery process. Besides, another study by *Ford et al. (1956)* concluded that radiation-induced chimeras can be cytologically identified through the presence of distinct cellular markers from both donor and recipient tissues. The study provided evidence that after radiation exposure, transplanted cells from a donor can proliferate and integrate into the recipient's body, leading to a chimeric state. Later, these findings confirmed the existence of transplantable multipotent stem cells in the BM of adult mice, which exhibited long-term (LT) hematopoietic repopulating activity. This discovery led to the hypothesis that the original cells responsible for this activity could be identified and quantified based on the types of mature cells they replenished in recipients who had undergone myeloablation. Since their functional characterization in 1961, HSCs have been the subject of extensive research that led to Till and McCulloch introducing the first *in vivo* clonal assay, known as the spleen colony-forming unit (CFU-S) assay (*Till & McCulloch, 1961*). This is also the first study that established the CFU-S assay as a tool to detect the presence of immature progenitor cells within mouse BM cells populations that can produce colonies of megakaryocytes, granulocytes, erythrocyte and macrophages in the spleen of irradiated animals (*Till & McCulloch, 1961*). Then, another study reported that CFU-S exhibits multilineage potential and self-renewal capacity as demonstrated by the ability of a single CFU-S to repopulate into more CFU-S (*Siminovitch, McCulloch & Till, 1963*). This assay was designed to evaluate the functional capacity of BM-derived hematopoietic progenitors at the single-cell level. By transplanting BM cells and analyzing the resulting cellular clusters in the spleen, the CFU-S assay demonstrated the self-renewal and clonal differentiation capabilities of the hematopoietic progenitor (*Baena, Manso & Forsberg, 2021*).

The hematopoietic system is finely regulated and significantly important in mammals. Mammalian hematopoiesis occurs at the marrow cavity of the skeletal system, which is where the HSCs and their progenitors reside. However, the marrow cavity's microenvironment can be challenging for scientists to directly explore and assess their function. As a substitution, *in vitro* assays have been developed to functionally evaluate HSPCs (*Purton & Scadden, 2007*). A variety of *in vitro* and *in vivo* assays have been created to identify and measure hematopoietic cells at different stages of differentiation (*Wognum et al., 2013*). The fastest method is the flow cytometric analysis, which is also the only method that can prospectively identify and isolate subpopulations of HSPCs with differential phenotypes. However, phenotyping analysis alone has limitations in providing information about the functionality of respectively identified HSPCs's subpopulations. Nevertheless, the phenotyping data acquired using flow cytometry has been reported to correlate with the ability to repopulate and is therefore widely accepted in the field (*Yilmaz, Kiel & Morrison, 2006*). In addition to the phenotyping studies using flow cytometry, another common *in vitro* assay for HSPCs studies is a short-term (ST) culture assay known as the colony-forming unit (CFU) assay. The CFUs assay is regarded as a potency test that allows the functional assessment and quantification of the HSPCs in both animal and

human systems. According to *Thompson et al. (2023)*, the hematopoietic CFUs assay is an indispensable tool for understanding and characterizing the regenerative capability of these essential stem cells. It provides valuable insights into the HSPCs' potency by measuring their ability to undergo cell proliferation and differentiation, making it a fundamental technique for advancing our understanding on the biological property of HSPCs (*Skific & Golemović, 2019*).

Generally, the CFUs assay involves the isolation of HSPCs from various sources, including BM, peripheral blood or cord blood in humans or from relevant tissues in animal models. Its primary objective is to evaluate the functional potency of these cells by observing their ability to form discrete cell clusters or colonies with distinct morphology when cultured in a semi solid medium supplemented with appropriate growth factors that supports their growth and differentiation (*Wognum & Szilvassy, 2015*). Each colony represents a population of hematopoietic cells derived from a single progenitor stem cell. While it has some limitations, ongoing innovations continue to make the assay more precise and adaptable for different research contexts, ensuring its enduring relevance in scientific inquiry. The CFUs assay has a wide range of applications, from basic research to clinical contexts and ongoing innovations are expanding its utility, making it an enduring and evolving technique in the field of HSPCs research.

Overall, this review aims to explore the fundamental discoveries concerning utilization of CFUs assay in the context of HSPCs. This is done by providing a historical overview of the CFUs assay for the study of clonal hematopoiesis involving the multilineage potential of HSPCs, and its use in various experimental models comprising of human, mice/rodents, zebrafish and induced pluripotent stem cells (iPSCs). In addition, this review uncovers research gaps, the future direction and prospective role of the CFUs assay in the clinical and basic sciences. This article is intended for audiences from various backgrounds such as students, educators, scientists as well as medical professionals by providing a concise and up to date summary of developments in the subject of HSCs and CFUs assay that could benefit academic, research and therapeutic purposes.

The structure of this article is organized in the following sequential orders: The section "Survey Methodology" details the search criteria employed to locate articles and references that were used to construct this review. "Hematopoiesis: A Mirror of Blood Cells Development" elaborates on the hematopoiesis and its maintenance as regulated by the roles of HSCs as well as lineage-specific hematopoietic progenitors. "The Past, Current and Future Direction of Colony-forming Units Assay in Hematopoietic Stem and Progenitor Cells Research" provides an in-depth of the role of the CFUs assay in the context of animal and human-derived HSPCs which reflects the evolution of various refined techniques used to study the clonogenic potential and differentiation capacities of these crucial cell populations. Meanwhile, "Overview of Colony-Forming Units Assay from Different Hematopoietic Stem Cell Sources" provides crucial information concerning the application of the CFUs assay using HSPCs from different biological sources. "The Impact of Colony-forming Units Assay in Stem Cell Research and Medical Applications" discusses the relationship of the CFUs assay in stem cells research and their impact in medical applications. The role of the CFUs assay in the stem cells industries is explored

in "Unlocking the Roles of Colony-forming Units Assay in Hematopoietic Stem and Progenitor Cells Industries". Last but not least, the conclusion and future remarks of this review are presented in "Conclusions and Future Remarks". It is also important to note that in this review, the use of term CFU-S is referring to the *in vivo* CFU assay for spleen-colony forming units, while the term CFUs is referring to the *in vitro* hematopoietic-colony forming units assay.

## SURVEY METHODOLOGY

Table S1 summarizes the article selection criteria used to construct this review. Briefly, journal databases, primarily Google Scholar, Scopus, PubMed and Web of Science, were used to research scholarly articles reviewed in this article. A literature search to collect information on the chosen topic was conducted across the listed databases from October 2023 until August 2024. The keywords used to search for these articles include "colony-forming unit", "hematopoietic stem and progenitor cells", "hematopoietic lineages" and "methylcellulose". The selection for inclusion criteria required the articles to be closely related to the utilization and evolution of CFUs assay in the analysis of HSPCs along with its role in clinical and basic sciences. Meanwhile, articles published in languages other than English, incomplete articles, those with unavailable full texts and unpublished results were excluded in this review. This approach ensured that the review was constructed from the most reliable and pertinent information available. The searches were not narrowed down based on specific criteria such as publishing date, authors, author affiliations, journals, or the impact factors of the journals. The quantitative studies offered measurable data from experimental, epidemiological and clinical research, uncovering observable trends in the use of CFUs assay within stem cell research and clinical applications. Meanwhile, the articles addressing qualitative studies offered perspectives on the issues and theories that underpin the use of the CFUs assay. Using the aforementioned methodology, this review was constructed based on 106 references that were comprised of 62 original research articles, 35 review articles, three webpages, four books and two theses. These listed references are pertinent to the historical and emerging evidences related to the CFUs assay and their fundamental role in HSPCs analysis involving various models.

### Hematopoiesis: a mirror of blood cells development

Stem cells are a distinctive group of cells found at all stages of life, capable of self-renewal and differentiating into various cell types. They play a central role in developmental biology and are essential for the repair processes that occur after injury or disease in the specific differentiated tissue they originate from *Kolios & Moodley (2013)*. Stem cells have unique characteristics that distinguish them from fully matured cells, enabling them to differentiate into various specialized cell types based on their potency, differentiation capability, and origin (*Poliwoda et al., 2022*). This differentiation process is influenced by the category of differentiation potency that a stem cell possesses. There are different categories of potency associated with stem cells namely totipotent, pluripotent, multipotent, oligopotent and unipotent (Fig. S1) (*Kalra & Tomar, 2014*). HSCs are regarded as multipotent stem cells that are starting to show a more restrictive differentiation capacity which is limited to cell

types from a single germ layer. This is indicated by the undifferentiated phenotype of HSCs that are capable of differentiating into common myeloid and common lymphoid precursor cells, followed by further differentiation into downstream lineage-specific and matured blood cells.

During embryonic development, blood cells originate from the yolk sac and later from the fetal liver and spleen. This early hematopoiesis is crucial for supplying the developing embryo with oxygen and nutrients. It mirrors the embryonic stages of development and contributes to the formation of the vascular system. In adults, hematopoiesis primarily occurs in the BM, particularly in the cavities of certain bones. This ongoing process reflects the body's need for a continuous supply of mature blood cells to maintain homeostasis (*Chen et al., 2020*). Hematopoiesis involves the organization of stem cells, progenitor cells and differentiated blood cells. This hierarchy mirrors the hierarchical stages of development in which undifferentiated cells gradually give rise to specialized cell types (*Wei & Frenette, 2018*).

The HSCs niche refers to a complex microenvironment where HSCs and multilineage HSPCs are located. They migrate from their site of origin to the BM, where they establish a niche for continued hematopoiesis (*Pinho & Frenette, 2019*). This migration process reflects the movement of cells during embryonic development as they populate different tissues and organs. The BM microenvironment or niche plays a crucial role in regulating hematopoiesis. Hematopoiesis exhibits plasticity, allowing the adaptation of blood cell production in response to changing physiological needs (*Yamashita et al., 2020*). In addition, hematopoiesis is highly regulated by a complex network of signaling molecules, with cytokines, hormones and growth factors playing crucial roles in influencing various stages of hematopoietic development (*Zhang & Lodish, 2008*). Cytokines are signaling proteins that mediate communication between cells and they act as key regulators of hematopoiesis by controlling the self-renewal, differentiation, and maturation of HSCs and lineage-committed hematopoietic progenitor cells into functionally mature blood cells in the niche area (*Pietras, 2017*). The influence of cytokines on hematopoiesis is multifaceted and context-dependent.

Hematopoiesis starts from the asymmetric division of HSCs, where they produce new stem cells through self-renewal and also differentiation to produce progenitor cells that are multipotent progenitors (MPP) (*Seita & Weissman, 2010*). The progenitor cells will further divide to produce two types of progenitor cells, which are the myeloid, CMP or lymphoid progenitors, CLP. The CMP will conduct cell differentiation to produce two types of progenitors, namely the myelomonocytic, megakaryocyte-erythrocyte progenitors (MEP) and the myelomonocytic, granulocyte-monocyte progenitors (GMP), which will form erythrocytes/platelets and granulocytes/macrophages, respectively. Meanwhile, the CLP functions in the production of Pro-T, Pro-B and Pro-NK type progenitors from which these three types of progenitors can form mature white blood cells such as T cells, NK cells and B cells, respectively (*Seita & Weissman, 2010*) (Fig. S2).

HSPCs are able to give rise to different types of blood cells through the CFUs assay. The most frequently utilized *in vitro* assays are also known as the colony-forming cell (CFC) assays for assessing the proliferative and differentiative capacity of HSPCs into differential

hematopoietic lineages (*Sarma, Takeda & Yaseen, 2010*). Technically, CFUs assays involve plating a single cell suspension of HSPCs at a designated cell density in a semi solid media, such as those based on methylcellulose with an addition of appropriate cytokines (*Ezeh et al., 2016*). Under these conditions, individual progenitor cells or CFUs are supported for proliferation and differentiation, leading to the formation of distinct colonies. These colonies, originating from various types of progenitor cells, are identified and tallied according to the number and kinds of mature cells they comprise, using morphological criteria (*Kronstein-Wiedemann & Tonn, 2019*). The CFUs assay is primarily employed to identify multipotential and lineage-restricted progenitors within the erythroid, granulocytic and macrophage lineages. It can also detect megakaryocyte and B-lymphoid progenitors when specific culture conditions tailored to these progenitors are used. While purified HSPCs are capable of forming colonies under suitable culture conditions, most CFUs found in the BM, blood, and other tissues consist of progenitors with limited self-renewal capacity and a constrained ability to repopulate hematopoietic cells *in vivo* (*Sawai et al., 2016*). However, the CFUs assay can act as a valuable substitute for HSPCs in situations where LT transplantation assays are either too costly or not feasible.

In essence, the CFUs assay can detect two types of erythroid progenitor cells namely the colony-forming unit-erythroid (CFU-E) and the burst-forming unit-erythroid (BFU-E) (*Dulmovits et al., 2017*). For the myeloid lineage, CFU-M (macrophage) and CFU-GM (granulocyte/macrophage) are differentiated based on the types of cells that make up the colonies they produce. Meanwhile the CFUs assay that is available for lymphoid progenitors is only limited for the detection of pre-B lymphoid progenitor cells (*Radtke et al., 2019*). An inverted light microscope can be used to observe colonies corresponding to lineage-specific HSPCs derived from CFUs assays. The morphological characteristics for each progenitor are described in Table S2 (*StemCell Technologies, 2017*). Since it was introduced more than forty years ago, the CFU assay has established itself as the standard *in vitro* functional test for investigating hematopoietic progenitor cells (HPCs). The CFUs assay is extensively utilized to examine the stimulating and inhibiting impacts of growth factors and to assess the effects of various *in vitro* interventions on cells. This includes processes such as cell handling, cryopreservation and gene transduction on cellular products employed in hematopoietic cell transplantation (*Bradley & Metcalf, 1966*). While more primitive HSCs mediate LT engraftment following transplantation, the quantity of CFUs in a graft has been linked to the efficiency of neutrophil and platelet engraftment and overall survival post-transplantation (*Page et al., 2011*). Therefore, the CFUs assay is a valuable tool for predicting the quality of a graft and has been especially effective in helping to select a higher number of functionally viable progenitor cells before HSPCs transplantation.

## The past, current and future direction of colony-forming units assay in hematopoietic stem and progenitor cells research

The increasing capability to cultivate mammalian cells *in vitro*, combined with established clonal assays, facilitated the establishment of protocols that led to the development of *in vitro* hematopoietic CFUs assays, initially for mice cells and subsequently for human cells. The history of the CFUs assay in the context of animal and human HSPCs reflects

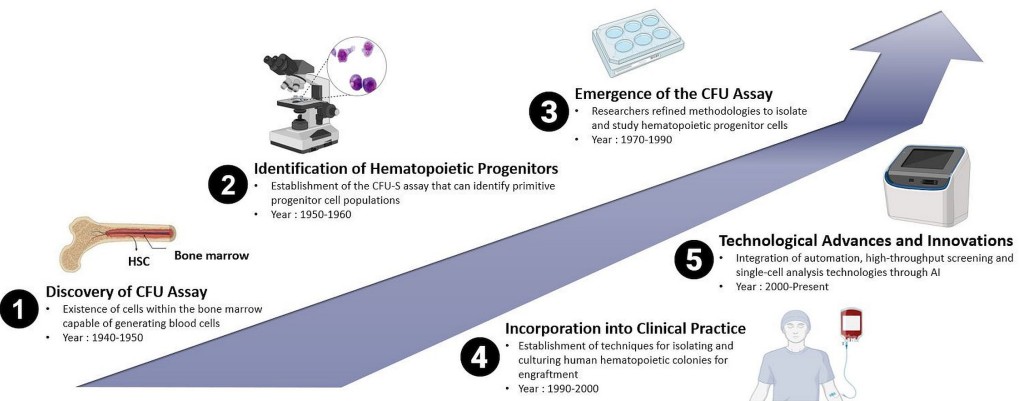

**Figure 1** Historical evolution of CFUs assay over time: from the discovery of the conventional method of CFUs assay to the deep learning approach-based CFUs assay driven by Artificial Intelligence (AI).

the evolution of techniques used to study the clonogenic potential and differentiation capacities of these crucial cell populations. While the concept of CFUs has been known for several decades, the methods have undergone refinement over time. Therefore, the history of the CFUs assay is marked by a series of milestones that have shaped its application in both animal and human HSPCs research. From the early identification of CFUs to the incorporation of advanced technologies, the CFUs assay continues to be a fundamental and evolving tool in the study of hematopoiesis as shown in Fig. 1. Research in this area of interest has been ongoing since the 1940s and remains a subject of active investigation in the current scientific world.

The groundwork for the CFUs assay was laid with early observations of HPCs. In the late 1940s, studies involving BM transplantation in animals were conducted, providing initial evidence for the existence of cells within the BM capable of generating blood cells. During the 1950s, researchers observed that mice exposed to whole-body irradiation showed improved survival rates when their spleens were externally protected (*Jacobson et al., 1951*). In these experiments, protection was provided to one of the legs or femurs, a portion of the liver, or the intestine. They noted that while this led to increased survival, it was less effective than when the spleen was shielded. This highlighted the critical role of the immune system and later identified key sites for the growth and sustenance of hematopoietic cells (*Jacobson, 1954*).

Later, another study demonstrated the viability of BM transplantations by successfully transplanting BM from a mouse donor into irradiated mice recipients. Subsequently, they observed the formation of small lumps in the spleen, which led to the early discovery of colonies containing cells, identified as HSCs (*Till & McCulloch, 1961*). Additionally, it was also demonstrated that the number of spleen colonies correlated with the number of transplanted BM cells, indicating the presence of a specific subset of hematopoietic cells with the ability to restore hematopoiesis. This was the first evidence that elucidated the migration ability of transplanted HSCs to migrate to a specific niche within the BM microenvironment in the recipient body. Migration is a biological process that is the

ability of the donor HSCs to migrate, engraft and proliferate in the recipient's body to restore defective hematopoiesis, with the BM being their final destination. Their repeated experiments aimed to examine the resilience of healthy BM cells to radiation and provided cytological evidence of interactions between the cells (*McCulloch & Till, 1962*; *Wu et al., 1968*). Through these continuous experiments, the researchers were able to demonstrate that stem cells derived from the spleen colonies in BM are capable of generating new colonies, confirming the regenerative capacity of a single cell through direct cytology (*Till & McCulloch, 1961*; *McCulloch & Till, 1962*; *Becker, McCulloch & Till, 1963*; *Siminovitch, McCulloch & Till, 1963*). Thus, it is a cornerstone in understanding cellular biology and has significant implications in various fields, particularly in regenerative medicine and stem cell research.

Moreover, in the 1970s and 1980s, the CFUs assay gained prominence with the development of techniques to culture and identify hematopoietic colonies. The focus was initially on murine models, and researchers refined methodologies to isolate and study HPCs. They identified different types of CFUs representing various stages of hematopoietic cell differentiation. These included CFU-GEMM (granulocyte, erythrocyte, monocyte, megakaryocyte), CFU-GM (granulocyte-macrophage), and others. This added complexity and specificity to the assay. CFU-GEMM is defined as the most primitive HPC with the unique potency to form multilineage colonies consisting of granulocytes, erythrocytes, monocytes, and macrophages in the presence of specific growth factors such as IL-3 (*Monette & Sigounas, 1987*). Meanwhile, erythroid-committed progenitors will form two distinct types of colonies namely the CFU-E and BFU-E. In the presence of erythropoietin (EPO), CFU-E divides rapidly and gives rise to single erythroblast colonies, whereas BFU-E is generally a slow-dividing cell that gives rise to larger colonies of erythroblasts (*Kimura et al., 1984*). The immature slow-dividing BFU-E differentiates into intermediate mature BFU-E and then to fast-dividing CFU-E cells (*Wu et al., 1995*). CFU-GM is the progenitor cell for GM and gives rise to single lineage-committed progenitors, CFU-G (granulocyte) and CFU-M (macrophage), under the influence of colony-stimulating factors (CSF) such as GM-CSF, G-CSF, and M-CSF (*Metcalf & Burgess, 1982*). The biological process of generation of megakaryocytes and platelets is known as megakaryocytopoiesis. CFUs of megakaryocytes (CFU-Mk or CFU-Meg) are unipotent in nature and in the presence of thrombopoietin (TPO) give rise to mature megakaryocytic cells (*Broudy et al., 1996*; *Kimura et al., 1984*).

As the understanding of hematopoiesis expanded, the CFUs assay was adapted for human HSPCs. This development saw the establishment of techniques for isolating and culturing human hematopoietic colonies, paving the way for clinical applications. From these experimental findings, the field of clinical hematology embraced the practice of HSC transplantation by using BM cells as the main source to obtain HSCs. The earliest BM transplants were allogeneic, intended to restore the hematopoietic system in humans exposed to radiation following a nuclear accident. However, over time, the preferred way to harvest HSPCs have shifted to emerging sources such as mobilized peripheral blood-derived HSPCs and cord blood cells (*Mathe et al., 1959*). The CFUs assay became integral to HSCs transplantation. In the 1990s, it was employed to assess the quality and

clonogenic potential of donor cells, ensuring successful engraftment. A study conducted by *Page et al. (2011)*, showed that CFUs dose is a strong independent predictor of engraftment after unrelated umbilical cord blood transplantation (UCBT) in human and should be used to assess potency when selecting CBUs for transplantation. Cord blood banking also emerged during this period, with the CFUs assay being used to evaluate the efficacy of stored cord blood units (*Skific & Golemović, 2019*).

Before discussing umbilical cord blood banking, it is essential to understand that human blood, BM, body tissues, skeletal muscles, and embryos are important stem cells sources (*Zakrzewski et al., 2019*). The composition of umbilical cord blood is known to be 40% monocytes and 40% lymphocytes and the remaining 20% are neutrophils and progenitor cells (*Theunissen & Verfaillie, 2005*). *In vitro* studies have shown that $CD34^+$ cells from umbilical cord blood (UCB) of human sample proliferate more rapidly than those from BM. Additionally, when UCB is transplanted *in vivo*, it demonstrates superior regulatory capabilities compared to BM stem cells (*Theunissen & Verfaillie, 2005*). With the help of CFUs assay, a defined seeding density can predict better cord blood potency.

Advances in cell biology, microscopy and molecular techniques have continuously improved the CFUs assay. Automation, high-throughput screening and single-cell analysis technologies have been integrated into the assay, enhancing its precision, efficiency and versatility. Counting cells and colonies *via* automated system is an integral part of high-throughput screens and quantitative cellular assays (*Choudhry, 2016*). Recent advancements in microfluidic technology have the potential to greatly enhance stem cell research by mimicking physiological environments to support the growth of stem cells. However, the use of microfluidic technology in CFUs application remains underreported which indicates the need for future exploration (*Zhang & Austin, 2012*). A critical method for evaluating their effectiveness and forecasting successful engraftment involves measuring the number and quality of lineage-specific progenitor cells and multipotent stem cells within the transplant population. Previously, a direct relationship between the quantity of HSPCs and engraftment success in human has been reported (*Page et al., 2011*). Commonly utilized indicators to determine the potency and quality of the graft include the total count of viable nucleated cells, the presence of $CD34^+$ antigen-expressing cells, and the ability of cells to form distinct colonies of mature blood cells in semi solid growth media (*Patterson et al., 2014*). The adoption of an automated method would enhance both the precision and speed, thereby improving standardization in conducting the CFUs assay and result analysis (*Pamphilon et al., 2013*).

Assessing the type of CFUs based on morphological scoring which is conducted manually under microscopic observation involves subjective judgments about the shape, size, and other visual characteristics of colonies, which can vary depending on the observer's experience and expertise. This variability can lead to human error and inconsistencies in data reporting, impacting the reliability of CFUs assessments. Hence, finding a new approach that could overcome this obstacle is essential to enhance accuracy and reproducibility in CFUs assay analysis, paving the way forward for successful experimental and therapeutic outcomes. One of the emerging technology-driven approaches that can be employed in CFUs assay is Artificial Intelligence (AI). By using AI-based technology,

algorithms can be trained to recognize and count CFUs colonies through image processing. This would improve accuracy, eliminate subjectivity, and significantly speed up the analysis process (*Kusumoto & Yuasa, 2019*). AI systems could be designed to identify irregularities or anomalies in colony formation patterns, thus could help researchers to quickly identify issues such as contamination or unexpected changes in cell behaviour. AI can assist in integrating data from various sources, such as genetic information, cell signaling pathways and environmental factors, providing a more comprehensive understanding of the factors influencing CFUs formation. According to *Kusumoto & Yuasa (2019)*, the primary domains within AI encompass computer vision, natural language processing (NLP), machine learning (ML), autonomous vehicles and robotics. ML is a type of algorithm that enables a computer to identify and categorize patterns within large data sets autonomously, without needing explicit programming for each task (*Kusumoto & Yuasa, 2019*). One specific method within ML, known as the convolutional neural network (CNN), falls under the category of supervised deep learning. CNNs have notably enhanced the accuracy of image recognition studies, demonstrating significant improvements in various research outcomes (*Zeng et al., 2016*). Deep learning-based approaches to analyze CFUs represent a significant advancement in the field of cell biology, particularly in the context of HSPCs research.

It has been demonstrated that AI technologies utilizing CNNs provide an effective platform for studying iPSCs, particularly in the analysis of colonies. As a result, the application of CNNs is poised to expand in the near future to encompass other types of stem cell analysis. HSCs could also benefit from these advancements in CNNs. Alongside iPSCs, HSCs represent one of the most established and widely utilized sources of stem cells for both medical and research purposes. HSCs play a critical role in maintaining the hematopoietic system (*Butko, Pouget & Traver, 2016*). They serve as a valuable resource for stem cell-based therapies and various hematological research studies. These studies span a wide range of applications, including toxicological assessments, developmental biology investigations, and drug testing endeavors (*Ng & Alexander, 2017*). Therefore, it is imperative to thoroughly investigate and validate the properties of HSCs using functional assays specific to HSCs.

Meanwhile, time-lapse microscopy allows for the continuous monitoring of colony growth, providing insights into dynamic changes in morphology over time. The capability to gather data on the quantifiable characteristics of progenitors as they form colonies offers a foundation for conducting mechanistic studies (*Scanlon et al., 2022*). For instance, this method can uncover the connection between cell cycle speed and fate specification as indicated by existing research, and it can assess the extent to which upstream progenitors engage in symmetric *versus* asymmetric self-renewal (*Berg, 2014*). Last but not least, quantitative morphological analysis such as digital pathology tools and image analysis software could be leveraged for quantitative analysis of hematopoietic CFU's morphology. This includes observation of parameters such as size, shape, and density of the colonies.

CFUs assay can offer relatively simple *in vitro* model and require less sophisticated equipment compared to *in vivo* models, making them accessible for many laboratories. However, CFUs assay has limitations and is unsuitable to completely replace *in vivo* models. While it is an excellent tool for preliminary testing and screening, *in vivo* models remain

essential for several reasons. *In vivo* models can assess both ST and LT HSPCs functionality, including self-renewal, differentiation, and engraftment. Besides, preclinical validation of therapies or drugs requires data from *in vivo* models to ensure safety and efficacy before advancing to clinical trials. Their models provide a more accurate representation of how cells behave in the context of a living organism, which is crucial for translational stem cell research. Thus, CFUs assay can only offer alternative and complementary models to study HSPCs in addition to *in vivo* models.

## Overview of colony-forming units assay from different hematopoietic stem cell sources

A fundamental step in stem cell research and various medical applications is HSCs isolation. HSCs are the precursor cells that give rise to all the different types of blood cells in the body, making them vital for various therapeutic approaches, including BM transplantation and regenerative medicine. The isolation process involves the separation of HSCs from other cell types in a sample, such as BM, peripheral blood or cord blood and must be carried out with precision and care to ensure the purity and functionality of the isolated HSCs, which are essential for successful clinical outcomes and scientific advancements in the field of stem cell research (*Morgan et al., 2017*). HSCs can be obtained from different sources, such as BM, peripheral blood, and umbilical cord blood in humans (*Lee & Hong, 2020*). In animal models, HSCs are typically isolated from specific tissues, depending on the species used. The choice of source often depends on the intended application and the availability of suitable samples.

Stem cells are primarily classified into three types including embryonic stem cells (ESCs), adult stem cells and iPSCs. Each type has unique characteristics and possible applications. Both iPSCs and the CFUs assay are important elements in stem cell research. For example, iPSCs have the ability to differentiate into various cell types, including hematopoietic cells. Researchers can induce iPSCs to undergo hematopoietic differentiation protocols to generate cells of the blood lineage, including HSPCs. Usage of adult stem cells and iPSCs have become increasingly significant in medical applications, particularly for the regeneration and repair of damaged tissues due to their better availability and less ethical concerns as compared to ESCs (*Seita & Weissman, 2010*). Prior to stem cells application in research and therapies, it is crucial to accurately determine and verify their potency and function. This process requires trained scientists or operators for precise analysis. To date, CFUs analysis is performed through a fundamental technique that requires the need for microscopic observation of the colonies. Despite its importance, these analyses are typically carried out using traditional techniques (*Ramakrishna et al., 2020*). For example, the functional characterization of HSCs and iPSCs involves the microscopic identification and classification of colony subtypes through CFUs assays. By observing these colonies under a microscope, scientists can classify them based on their morphological characteristics, which reflect the types of cells they can differentiate into. This conventional method, which must be performed by a trained operator, is not only labor-intensive and time-consuming but also susceptible to inaccuracies in data reporting.

Prior to the CFUs assay, various techniques have been employed in sample preparation. For instance, the sample containing HSCs is collected and processed to remove any debris, red blood cells, and contaminants. This preparation step is crucial to obtain a pure population of HSCs. Depending on the specific application, the isolated HSCs may be cultured and expanded *in vitro* to obtain a larger population of cells for transplantation or research purposes. In many cases, a density gradient centrifugation is used to separate the mononuclear cell fraction, that contains HSCs from other blood components (*Pösel et al., 2012*). This technique relies on the differences in density among blood cell types, allowing the HSC-containing fraction to be isolated. Immunoselection is another common method for HSC isolation. It involves the use of specific antibodies that target unique cell surface markers expressed on HSCs (*Jaatinen & Laine, 2007*). By binding to these markers, the HSCs can be selectively separated from other cells using techniques such as magnetic-activated cell sorting (MACS) or fluorescence-activated cell sorting (FACS) (*Bacon et al., 2020*). While FACS offers several benefits, including faster analysis and the ability to assess multiple parameters simultaneously, its drawbacks include the need for specialized equipment that is costly and the requirement for cells to be modified to exhibit a fluorescent feature (*Shen et al., 2021*). On the other hand, MACS avoids these drawbacks as it does not require specialist equipment or fluorescent labels. Instead, MACS utilizes magnetic particles that are functionalized to bind to specific cell subsets in a mixture, thereby enabling separation (*Schmitz et al., 1994*). The isolated HSCs are typically subjected to rigorous quality control measures to ensure their purity, viability and functionality. This may involve testing their differentiation ability and genetic integrity.

An overview of CFUs for various adult HSPCs models derived from humans and rodent/mice is shown in Table 1A. In addition, Table 1B addresses the CFUs for fetal sources from humans and rodent/mice while, Table 1C shows CFUs from other sources such as zebrafish and iPSCs. *In vitro* CFUs assays are commonly utilized in numerous studies to assess the quantitative and qualitative characteristics of HSCs. For example, *Li et al. (2016)* used the CFUs assay on BM samples from 365 patients that were recently diagnosed with myelodysplastic syndrome (MDS). The findings indicated that abnormalities in the proliferation and differentiation of erythroid and myeloid precursor cells *in vitro* mirror the ineffective hematopoiesis characteristic of MDS. These results could be valuable in forecasting outcomes for individuals with higher-risk MDS. A notable study in this area by *Shpall et al. (2002)*, investigated the feasibility of using *ex vivo* expanded umbilical cord blood for transplantation. They utilized CFU assays to evaluate the growth and differentiation ability of cord blood-derived HSCs following *ex vivo* expansion, demonstrating the clinical utility of expanded HSCs in enhancing engraftment and reducing the time to hematopoietic recovery in human transplant recipients (*Shpall et al., 2002*).

Several studies in the 1990s used CFUs assays to show that embryos of mice exposed to benzene during pregnancy exhibited changes in the number of progenitor and hematopoietic precursor cells, along with an increase in granulopoiesis (*Corti & Snyder, 1996*; *Keller & Snyder, 1988*). More recently, our research has identified a lineage-specific response to benzene toxicity among HSPCs by using the CFUs assays (*Chow et al., 2015*;
**Table 1  Overview of CFUs from various model.**

| Study model | Sample collection | Sample processing method | Growth media and culture conditions | References |
|---|---|---|---|---|
| **A. Adult** | | | | |
| Human | Bone marrow | Ficoll-Hypaque gradient density centrifugation | • Type of culture: Suspension<br>• Growth media: DMEM<br>• Semi solid media: Methylcellulose agar (MethoCult™ (cat. no: H4435))<br>• Culture condition: Incubator at 37 °C with 5% $CO_2$ and >85% humidity | *Li et al. (2016)* |
| | Cord blood | Centrifugation | • Type of culture: Suspension<br>• Growth media: DMEM<br>• Semi solid media: Methyl-cellulose agar (Amgen)<br>• Culture condition: Not mentioned | *Shpall et al. (2002)* |
| Pregnant Swiss webstermice | Bone marrow | Flushing technique | • Type of culture: Suspension<br>• Growth media: Alpha<br>• Semi solid media: Methylcellulose agar (MethoCult™ (cat. no: #03334))<br>• Culture condition: Humidified incubator at 37 °C with 5% $CO_2$ | *Corti & Snyder (1996)* |
| ICR mice | Bone marrow | Flushing technique | • Type of culture: Suspension<br>• Growth media: DMEM<br>• Semi solid media: Methylcellulose agar (MethoCult™ (cat. no: #03534, #03334, #03630))<br>• Culture condition: Humidified incubator at 37 °C with 5% $CO_2$ | *Chow et al. (2015)* and *Chow et al. (2018)*<br>*Yi et al. (2018)*<br>*Mohd Idris et al. (2018)*<br>*Abd Hamid et al. (2020)*<br>*Dewi et al. (2021)* |
| Cyp1b1-null and wild type mice | Bone marrow | Flushing technique | • Type of culture: Suspension<br>• Growth media: RPMI<br>• Semi solid media: Methylcellulose agar (MethoCult™ (cat. no: #03534, #03630))<br>• Culture condition: Humidified incubator at 37 °C with 5% $CO_2$ | *N'jai et al. (2010)* |
| Ts65Dn and euploid mice | Bone marrow | Not mentioned | • Type of culture: Suspension<br>• Growth media: Not mentioned<br>• Semi solid media: Methylcellulose agar (Methocult™ (cat. no. MC3534, MC3630))<br>• Culture condition: Not mentioned | *Lorenzo et al. (2011)* |
| **B. Fetal** | | | | |
| Human | Umbilical cord blood | Centrifugation | • Type of culture: Suspension<br>• Growth media: serum-free medium (Stem$\alpha$)<br>• Semi solid media: Methylcellulose agar (Stem$\alpha$)<br>• Culture condition: Not mentioned | *Ivanovic et al. (2004)* |

**Table 1** (*continued*)

| | | | | |
|---|---|---|---|---|
| Pregnant CD-1 and C57B1/6N mice | Fetal liver | Flushing technique | • Type of culture: Suspension<br>• Growth media: Iscove's modified Dulbecco's<br>• Semi solid media: Methylcellulose agar (MethoCult™ (cat. no: M3334, M3534))<br>• Culture condition: Incubator at 37 °C with 5% $CO_2$ | *Badham & Winn (2010)* |
| Pregnant CD-1 and C57B1/6N mice | Fetal liver | Flushing technique | • Type of culture: Suspension<br>• Growth media: Iscove's modified Dulbecco's<br>• Semi solid media: Methylcellulose agar (MethoCult™ (cat. no: M3334, M3534))<br>• Culture condition: Incubator at 37 °C with 5% $CO_2$ | *Badham et al. (2010)* |
| Pregnant Swiss webstermice | Fetal liver | Mincing technique | • Type of culture: Suspension<br>• Growth media: Supplemented alpha<br>• Semi solid media: Methylcellulose agar (MethoCult™ (cat. no: #03334))<br>• Culture condition: Humidified incubator at 37 °C with 5% $CO_2$ | *Corti & Snyder (1996)* |
| Pregnant Swiss webstermice | Fetal spleen and liver | •Aspirations using a siliconized fine-tipped Pasteur pipet.<br>• Suspensions of splenic cells were obtained by pressing via wire mesh grid | • Type of culture: Suspension<br>• Growth media: Supplemented alpha<br>• Semi solid media: Methyl-cellulose agar: Not mentioned<br>• Culture condition: Humidified incubator at 37 °C with 5% $CO_2$ | *Keller & Snyder (1986)* |
| Pregnant SPF Kunming mice | Yolk-sac | Pipetting up and down | • Type of culture: Suspension<br>• Growth media: StemSpan™ Serum-Free Expansion<br>• Semi solid media: Methyl-cellulose agar (MethoCult®)<br>• Culture condition: Humidified incubator at 37 °C with 5% $CO_2$ | *Zhu et al. (2013)* |
| **C. Other sources** | | | | |
| Zebrafish | Embryo | Not mentioned | • Type of culture: Suspension<br>• Growth media: embryo medium<br>• Semi solid media: Methyl-cellulose agar (not mentioned)<br>• Culture condition: Not mentioned | *Traver et al. (2003)* |
| Zebrafish | Fetal liver | Flushing technique | • Type of culture: Suspension<br>• Growth media: not mentioned<br>• Semi solid media: Not mentioned<br>• Culture condition: Incubator at 37 °C with 5% $CO_2$ | *He et al. (2015)* |

**Table 1 (*continued*)**

| iPSCs | iPSCs-derived HSCs | Flushing and mincing techniques | • Type of culture: Suspension<br>• Growth media: DMEM<br>○Semi solid media: Methyl-cellulose agar (S-clone SF-O3)<br>• Culture condition: Incubator at 37 °C with 5% $CO_2$ | *Suzuki et al. (2013)* |

*Chow et al., 2018*). This research indicates that exposure to 1,4-benzoquinone (1,4-BQ) selectively reduces the clonogenicity of myeloid progenitors compared to lymphoid progenitors, highlighting the importance of lineage-specific mechanisms in the impact of benzene toxicity on the HSPC niche. Moreover, several other benzene metabolites studies also used the CFUs assay to form HSPCs colonies (*Yi et al., 2018*; *Mohd Idris et al., 2018*; *Abd Hamid et al., 2020*; *Dewi et al., 2021*).

A previous study by *N'jai et al. (2010)* used CFUs assay to demonstrate the rapid suppression (within 6 h) of lymphoid (CFU-PreB) and myeloid (CFU-GM) progenitor cells in 7,12-dimethylbenzathracene (DMBA) treated mice. The changes were not seen in Cyp1b1 null mice, indicating that local metabolism of DMBA in the BM by Cyp1b1 is necessary to affect BM CFU-PreB and CFU-GM. Additionally, these findings suggest that myeloid lineage cells recover more rapidly than lymphoid lineage cells following exposure to DMBA (*N'jai et al., 2010*). Meanwhile, the function of lymphoid progenitor cell in Down syndrome was observed using the CFUs assay. Examination of hematopoietic progenitor populations revealed that Ts65Dn mice had a reduced number of functional HSCs and a notably lower percentage of BM lymphoid progenitors (*Lorenzo et al., 2011*).

A number of studies have used the CFUs assays to grow fetal HSPCs. A notable study in this context is *Ivanovic et al. (2004)*, who investigated the characteristics of HSCs within the early human embryo, focusing on their expression profiles and functional capacities. The study utilized CFUs assays to evaluate the clonogenic potential of isolated HSCs from fetal umbilical cord blood, demonstrating their ability to give rise to various hematopoietic lineages. Previous studies have shown that transplacental exposure to benzene in animal models such as C57BL/6N mice, increases reactive oxygen species (ROS) level in the fetal liver and finally affecting the colony counts of erythroid and myeloid progenitors (*Badham & Winn, 2010*; *Badham et al., 2010*). Meanwhile, exposing pregnant mice to benzene from gestation day (GD) 6 to 15 leads to persistent changes in fetal tissue hematopoiesis that last up to six weeks post-birth, as evidenced by the disrupted development of myeloid and erythroid progenitor cells (*Keller & Snyder, 1986*). Earlier research involving fetal samples revealed that HSCs derived from the yolk sac are more vulnerable to the cytotoxic effects of hydroquinone (HQ) compared to HSCs derived from adult BM when culturing using semi solid methylcellulose agar (*Zhu et al., 2013*).

Other sources of stem cells that have gained interest in relation to the usage of the CFUs assays are from zebrafish and iPSCs. Researchers have created a clonal, ST *in vitro* assay for zebrafish erythroid and myeloid progenitor cells using methylcellulose culture methods. This assay supported the growth of zebrafish hematopoietic progenitor cells and allowed for the quantification of progenitor numbers in adults, enhancing the understanding of

HSPCs in zebrafish (*Traver et al., 2003*). Additionally, a study by *He et al. (2015)* who performed CFU-C assays on fetal liver-derived HSCs from Nlrc3$^{-/-}$ embryos of zebrafish, revealed compromised colony formation abilities. Meanwhile, research involving the use of the CFUs assay in iPSC-derived HSC is a critical area within regenerative medicine and stem cell biology. Researchers have developed a novel method for generating functional HSCs from mouse iPSCs. CFUs assays were utilized among other techniques to evaluate the hematopoietic capacity of the derived cells, demonstrating their ability to differentiate into various blood lineages and reconstitute the hematopoietic system in recipient mice (*Suzuki et al., 2013*).

As discussed above, there are various HSCs sources and experimental models that require application of the CFUs assay for the study of HSCs and hematopoiesis. However, further exploration is required to establish a standard and optimal protocol of CFUs assay for the respective models and sources. Ongoing research on refining the CFUs protocol is fundamental as the outcome can improve and expand its application for various HSCs sources and experimental needs. The following section of this review will delve deeper into the impact of CFUs in both stem cell research and how it affects medical applications.

## The impact of colony-forming units assay in stem cell research and medical applications

The CFUs assay plays a crucial role in stem cell research as it is a valuable tool for assessing the functional properties of stem cells. The CFUs assay is a powerful tool in stem cell research for assessing the lineage potential of individual stem and progenitor cells (*Thompson et al., 2023*). The CFUs assay has been used to study HSCs since its development in the 1960s, providing insights into their self-renewal, clonal differentiation and multilineage potential (*Weissman & Shizuru, 2014*). It has also been adapted to acquire data from single purified HSC populations, further advancing our understanding of the hematopoietic hierarchy (*Skific & Golemović, 2019*). The CFUs assay is considered the gold standard for assessing the potency of stem cell products, as it correlates with engraftment success and overall survival in recipients of HSCs transplantation (*Velier et al., 2019*). In the following sub-section, the impact of the CFUs assay in HSPCs analysis in relation to stem cells research and therapeutic application will be further described.

### *Identifying HSC subpopulations*

The CFUs assays have been instrumental in characterizing distinct subpopulations of HSCs based on their differentiation potential. Numerous studies have used CFUs assays to differentiate between LT and ST repopulating HSCs and their functional properties. LT-HSCs sustain life-long hematopoiesis and are characterized by a delayed engraftment pattern upon transplantation, while ST-HSCs provide early and transient hematopoietic recovery in human (*Notta et al., 2011*). A previous study used the CFUs assay alongside NS-GFP transgene expression to dissect mice HSC populations. By analyzing LSK (Lin$-$Sca-1$^{+}$c-Kit$^{+}$) cells divided into fractions based on NS-GFP intensity, the study identified subpopulations with varying degrees of stemness, including LT-HSCs (*Ali et al., 2017*). Researchers used the CFUs assay to assess the ability of stem cells to differentiate into various blood cell lineages, providing critical insights into hematopoiesis (*Guo et*

*al., 2017*). CFUs assays are designed to assess the colony-forming potential of specific cell lineages (*e.g.*, CFU-GM for granulocyte-macrophage lineage, CFU-E for erythroid lineages) thus they have allowed researchers to dissect the heterogeneity within stem cell populations. Differentiated colonies produced in these assays provide insights into the lineage commitment and differentiation ability of individual stem cells.

According to *Scanlon et al. (2022)*, the CFUs assay enables the quantification of progenitors committed to specific lineages. This has revealed the presence of distinct subpopulations of stem cells in human with preferences for generating particular cell types, contributing to the understanding of cellular diversity within stem cell populations. The CFUs assay can be adapted to single-cell cloning techniques, where individual cells are plated to form colonies. This approach allows the tracking of clonal dynamics, revealing the behaviour of single stem cells in terms of self-renewal and differentiation potential (*Cordes, Wu & Dunbar, 2021*). By analyzing the size and composition of colonies derived from single cells over time, researchers can study the clonal expansion and differentiation patterns of stem cells. This information is crucial for understanding the kinetics of stem cell divisions and the factors influencing clonal dynamics (*Upadhaya et al., 2018*).

The CFUs assay provides a quantitative measure of the frequency of colonies within a population. Rare cell populations with unique characteristics, such as LT repopulating potential or resistance to certain conditions, can be identified based on their colony-forming capacity (*Schreier & Triampo, 2020*). Researchers can use the CFUs assay to isolate and characterize rare subpopulations based on distinct colony types or behaviors, thus leading to the identification of cells with specific functional properties, such as the identification of quiescent or dormant stem cells that may play a role in tissue regeneration (*Goodell, Nguyen & Shroyer, 2015*). Moreover, the CFUs assays have revealed functional heterogeneity within colonies, even when derived from a seemingly homogeneous population of stem cells. This underscores the dynamic nature of stem cell populations and the importance of considering functional diversity when studying their behaviours. The study of CFU assay-derived colonies has provided valuable insights into the factors influencing stem cell fate decisions (*Quesenberry et al., 2022*). By understanding the heterogeneity in colony composition and behaviour, researchers can unravel the complex regulatory mechanisms governing stem cell fate determination. Therefore, the CFUs assay has significantly contributed to our understanding of stem cell heterogeneity, clonal dynamics, and the identification of rare cell populations (*Quesenberry et al., 2022*).

In addition, researchers are able to assess the morphology and composition of individual colonies by using this CFUs assay (*Skific & Golemović, 2019*). Variations in colony size, cell types present, and other characteristics provide insights into the functional heterogeneity of stem cells within a population. Researchers can also perform additional functional assays within individual colonies derived from CFUs assay. This includes gene expression analysis, immunophenotyping, and other techniques to understand the functional differences and molecular signatures within distinct colony types (*Wilson et al., 2015*). In conclusion, CFUs assay provides a versatile platform for studying the functional properties of stem cells and have paved the way for advancements in regenerative medicine and therapeutic applications.

### Therapeutic applications and disease modelling

It is undeniable that the CFUs assay has been a valuable tool for testing new drugs and evaluating potential therapies for hematological disorders. Examples of how this assay has been employed in preclinical and clinical studies to assess the effects of various compounds on HSCs function are given in this sub-section. HSCs are regarded as a vital cell source for treating regenerative diseases. The use of adult-derived HSCs in therapeutic settings is particularly versatile, establishing them as an essential resource in the field of stem cell biology. In cancer research, CFUs assays are used to study the clonogenic potential of cancer cells, aiding in the assessment of drug efficacy and resistance mechanisms (*Brix et al., 2021*). Meanwhile, *Bensinger et al. (2001)* used CFUs assay to evaluate the engraftment potential of HSCs in autologous transplantation settings. Their findings demonstrated a correlation between CFUs numbers and successful engraftment in humans, which informed protocols for stem cell harvesting and transplantation. A study by *Shpall et al. (2002)* demonstrated that the number of CFUs in cord blood units could predict neutrophil and platelet recovery post-transplant, leading to improved selection criteria for cord blood units in hematopoietic stem cell transplantation (HSCT) in human. Furthermore, an *ex vivo* expansion of UCB stem cells using CFUs assays have been done in humans to assess the proliferation and differentiation potential of expanded cells (*Delaney et al., 2010*). Their work has led to improved outcomes in cord blood transplantation, especially in adult patients, by enhancing the graft's cellular content. CFUs assays have been used to predict the engraftment likelihood and survival rates in HSCT.

Besides, the CFUs assay has been crucial in assessing the impact of genetic modifications and disease modelling on HSC function in both animal and human systems. The assay helps in characterizing the proliferation and differentiation capabilities of stem and progenitor cells under disease-specific conditions. Researchers use the CFUs assay to model various types of leukemia by introducing specific genetic mutations associated with leukemic transformation. This allows for the investigation of altered clonogenic potential, abnormal lineage commitment, and drug responses in leukemia-initiating cells (*Camacho et al., 2017*). CFUs assays are also employed to model aplastic anemia, a condition characterized by BM failure. The assay helps assess the decreased clonogenic potential of HPCs in this disease, contributing to the understanding of the disease pathophysiology. In MDS, which is a group of disorders characterized by ineffective blood cell production, the CFUs assay can be used to study the abnormal differentiation and reduced clonogenic potential of HPCs in humans (*Li et al., 2016*). Meanwhile in sickle cell disease research, the use of gene therapy was demonstrated to correct the sickle cell mutation in HSCs, with CFUs assay being used to assess the functionality of corrected cells, leading to promising clinical trials (*Ribeil et al., 2017*).

## Unlocking the roles of colony-forming units assay in hematopoietic stem and progenitor cells industries

Stem cells have gathered significant interest and applications in various industries due to their unique ability to differentiate into different cell types and their suitability for regenerative medicine. The hematopoietic CFUs assay offers applications not only in

research and clinical settings but also in the industrial context. Pharmaceutical companies use the CFUs assay to screen possible drug candidates that may affect HSPCs function. This is particularly relevant for drugs targeting hematological disorders or those with likely side effects on blood cell production. Besides, the CFUs assay helps to assess the potential toxicity of new compounds or industrial chemicals on HSCs, providing crucial safety information. Figure S3 shows a list of countries in relation to stem cell industries development up until 2020 (*World Population Review, 2024*). The United States leads with the highest number of trials, followed by South Korea, Australia and other countries, showcasing their respective contributions to the stem cell industry.

Companies involved in stem cell banking, including cord blood banks, use the CFUs assay to evaluate the quality of stored HSC samples. It helps ensure the clonogenic potential and viability of cells before storage or transplantation (*Pamphilon et al., 2013*). Industries focused on cell therapy, regenerative medicine and HSC-based therapies use the CFUs assay as a tool for assessing the potency and quality of therapeutic cell products, underscoring the importance of clonogenic potential in ensuring therapeutic efficacy (*Yüksel et al., 2010*). Furthermore, a study by *Fernández-Santos et al. (2022)* explored the optimization of cell culture conditions for HSCs using the CFUs assay. This research focused on the effects of various growth factors on the clonogenic potential of these cells, contributing to the development of more effective culture strategies.

Last but not least, ethical considerations are always a priority in stem cell research. The use of adult HSPCs instead of ESCs alleviates most ethical concerns that might otherwise arise. Adult HSPCs are typically derived from ethically non-controversial sources, such as BM, UCB, or mobilized peripheral blood. These sources do not involve the destruction of embryos or the creation of embryos specifically for research purposes, unlike ESCs (*Seita & Weissman, 2010*). The use of adult HSPCs reflects a commitment to advancing science while respecting ethical boundaries. It demonstrates that cutting-edge research can progress without crossing contentious ethical lines, showcasing a balance between innovation and responsibility.

## CONCLUSIONS AND FUTURE REMARKS

In conclusion, the CFUs assay has been pivotal in advancing our understanding of HSPCs, serving as a fundamental tool in stem cell biology and regenerative medicine. By enabling the assessment of the clonogenic potential and differentiation capacity of HSPCs, CFUs assays have facilitated key insights into the dynamics of hematopoiesis, the hierarchical organization of stem cells, and their lineage commitments. This assay has not only enhanced our basic scientific knowledge but was also translated into significant clinical advancements, particularly in the optimization of stem cell transplantation and the development of novel cell-based therapies for a range of hematological disorders. As we look to the future, the CFUs assay is expected to remain an indispensable part of stem cell research, adapting to and integrating with emerging technologies such as gene editing, single-cell sequencing and advanced imaging techniques. The integration of microfluidics and high-throughput analysis into CFUs testing represents a significant advancement in stem cell research. These

integrations promise to refine our understanding of HSPCs biology at an unprecedented resolution, enabling more precise manipulation of stem cell fate and function. Moreover, the expansion of CFUs assay applications beyond traditional models to include 3D culture systems, organoids, and even bioengineered tissues opens new avenues for exploring hematopoiesis in more physiologically relevant contexts. This evolution will likely lead to more sophisticated models of human diseases and more effective regenerative therapies. In the context of personalized medicine, leveraging CFUs assays in conjunction with genetic and molecular profiling could tailor therapies to individual patient needs, enhancing the efficacy and safety of stem cell-based treatments. Additionally, AI is increasingly being applied in various areas of stem cell research, with significant potential to transform the field. In stem cells like iPSCs and mesenchymal stem cells (MSCs), AI is used to analyze large datasets, predict cell differentiation, and optimize culture conditions, enhancing the accuracy and efficiency of experiments. For instance, a previous study has reported that deep learning models are able to analyze complex cellular phenotypes in a large-scale screen of various stem cell types. The AI model was able to identify subtle changes in cell morphology that corresponded to different biological outcomes. Therefore, by building on these successes, the application of AI in HSPCs research, particularly in CFUs analysis, shows great promise. AI can automate the counting and classification of CFUs, improve precision, and offer deeper insights into HSC behavior. This innovative approach could advance HSC research, aligning with current technological demands and enhancing its applications in regenerative medicine. By integrating with cutting-edge technologies and exploring new contexts, the CFUs assay continues to be a vital tool in unlocking the mysteries of stem cell biology and in advancing the frontiers of medicine. The ongoing evolution of CFUs assays heralds a new era of research and therapy, where their application could lead to breakthroughs in personalized medicine, drug discovery and tissue regeneration.

### Funding

This work was funded by the Fundamental Research Grant Scheme (FRGS), FRGS/1/2021/SKK06/UKM/03/1 from the Ministry of Higher Education (MOHE), Malaysia. This work was also supported by the Center for Diagnostic, Therapeutic and Investigative Studies (CODTIS), Faculty of Health Sciences, UKM. The funders had no role in study design, data collection and analysis, decision to publish, or preparation of the manuscript.

### Grant Disclosures

The following grant information was disclosed by the authors:
The Fundamental Research Grant Scheme (FRGS), FRGS/1/2021/SKK06/UKM/03/1 from the Ministry of Higher Education (MOHE), Malaysia.
The Center for Diagnostic, Therapeutic and Investigative Studies (CODTIS).
Faculty of Health Sciences, UKM.

## Competing Interests

The authors declare there are no competing interests.

## Author Contributions

- Nur Afizah Yusoff conceived and designed the experiments, performed the experiments, analyzed the data, prepared figures and/or tables, authored or reviewed drafts of the article, and approved the final draft.
- Zariyantey Abd Hamid conceived and designed the experiments, authored or reviewed drafts of the article, and approved the final draft.
- Siti Balkis Budin conceived and designed the experiments, authored or reviewed drafts of the article, and approved the final draft.
- Izatus Shima Taib conceived and designed the experiments, authored or reviewed drafts of the article, and approved the final draft.

## Data Availability

This is a literature review.

## Supplemental Information

Supplemental information for this article can be found online at http://dx.doi.org/10.7717/peerj.18854#supplemental-information.

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
