# Peer review of "Hematopoietic stem cell discovery: unveiling the historical and future perspective of colony-forming units assay"

_PeerJ, doi:10.7717/peerj.18854_

## Round 0.1 · original submission · Major Revisions

Dear authors,

As evident from the reviewers' reports, there are conflicting perspectives, but the misinterpretation or misquotation of references is particularly concerning. Accurate discussion of references is crucial in a review, as you should be well aware. So is clear definitions and use of the central concepts of your review. I am giving you the opportunity to revise the manuscript, but ensure that all claims are meticulously cited or referenced, rather than relying on random keyword searches online.
Thank you.

Reviewer 1 ·

Basic reporting

Unfortunately, I had to shorten the revewing at the end of the introduction. There are already for this part
5 citations (quoted in the text) missing from the bibliography (Weiskopf, Jacobson, Ford, Wognum, Sama)
3 citations with some mistakes (line 65: it s JACOBSON not Jacobsen (JACOBSON et al., Recovery from Radiation Injury, science, 1951 ; line 56 Pietras et al., is a review on HSC cycle only, not properly referenced ; line 94 Skific & Golemovic, 2019 are quoted 3 times, but one in 2017 without citation in the bibliography)
line 70: Till and McCulloch, not referenced : Till JE, McCulloch EA. A direct measurement of the radiation sensitivity of normal mouse bone marrow cells. Radiat Res 1961;178:Av3–7.

line 61-65: the interpretation is incorrect : text from JACOBSON et al., Recovery from Radiation Injury, science, 1951: “the factor responsible for recovery from radiation under these conditions is a substance of a noncellular nature. Itseems unlikely that(1)cells migrate out from the shielded or transplanted tissue and are responsible for the enhancement of recovery, or (2) that irradiation of tissue produces a"toxin"and that the shielded or implanted tissues exert a direct detoxifying action upon contact with the"toxin."
It would be necessary to take up all the references chosen to write the text, as it is not possible to review it in its entirety.

Experimental design

see Basic reporting

Validity of the findings

see Basic reporting

Additional comments

see Basic reporting

Reviewer 2 ·

Basic reporting

Yusoff et al. have provided a thorough review of the current uses of colony-forming unit (CFU) assays in hematopoietic stem cell research. The review is well-researched, and the authors clearly invested significant effort in compiling the relevant literature.
-The English language should be revised to ensure proper grammar, sentence structure, and smooth flow of paragraphs. A language revision by a colleague or a professional service could strengthen the manuscript.
-One of the key areas for improvement is the clarity of the writing. The review covers a wide range of topics, but organizing the content more clearly could make it more reader-friendly.
-The introduction could benefit from the inclusion of additional citations from older foundational articles that shaped our current understanding of hematopoiesis. Referencing these seminal works could provide more context for readers and anchor the review in the broader historical framework of the field.
-Figures and Tables:
--Figure 2: Please note that LT-HSCs should be placed higher in the hierarchy than ST-HSCs in the figure.
--Figure 4: This figure is not referenced in the text, and a mention of it should be added where appropriate.
--Table 3: Since iPSCs and zebrafish models are mentioned in the review, it would be helpful to include information about these models in the table to ensure consistency and provide readers with a comprehensive overview.

Experimental design

-Survey Methodology is clear and sufficient
-Primary and review articles are cited appropriately
-An opportunity to further enrich the review could be a more detailed exploration of in vitro CFU assays used in the field. For example, are the same assays with identical protocols and reagents used for murine and human cells? How do protocols change when the cell source shifts from HSCs to iPSCs or cancer cells?
-Introducing subheadings could help with organization. For instance, the section titled “The past, current and future direction of colony-forming unit assay in hematopoietic stem and progenitor cells research” could be divided into separate subsections for "Past," "Current," and "Future Directions." Similarly, the section on “Overview of colony-forming unit assay from different hematopoietic stem cell sources” could benefit from distinct subsections for HSCs and other cell sources.
-Throughout the manuscript, it is sometimes unclear whether the authors are referring to in vivo CFU-S assays or in vitro CFU assays. Clarifying this distinction would enhance the focus of the review.
-Avoid repeating the same information in multiple sections (e.g., the lineage potential of hematopoietic stem cells).

Validity of the findings

-Not in the conclusion, but in the section “The past, current and future direction" there is a really good argument for the implementation of AI in the analysis of these assays. This could be expanded on in the conclusion and a gap/future direction

---

## Round 0.2 · Minor Revisions

Dear authors, thank you for your revisions and wait. Both reviewers acknowledge the revision efforts and have a few minor points to add that I think that make sense, for the most part. Please, refer to the reviewers' reports for further details.

Reviewer 1 ·

Basic reporting

After examination, the synthesis is quite comprehensive regarding the history and perspectives of colony-forming unit (CFU) assays for hematopoietic stem cells. However, there may be some additional aspects that could further enrich this review, such as recent clinical applications of CFUs in the diagnosis and monitoring of hematological diseases, as well as the integration of new technologies like microfluidics or high-throughput analysis into CFU testing. Recent efforts to standardize CFU protocols on an international scale could also be discussed, along with ethical considerations related to the use of CFU tests, particularly in the context of embryonic stem cell research.

Experimental design

While these elements might be lacking, it is important to note that the synthesis already appears to be very complete and well-structured, covering historical, methodological, and application aspects of CFU tests in hematopoietic stem cell research.

Validity of the findings

not applicable

Additional comments

few minor points:
• Supplementary Figure S1: The term "leukocyte" is attributed to all white blood cells; the authors may have meant "myelocytes."
• Supplementary Figure S2: It might be interesting to add the different CFU colonies.
• Supplementary Figure S3: Could you specify a time scale for the number of trials? It might also be clearer to have the countries listed below each bar to avoid using similar colors.
• Lines 201-202 of the text: The term "homing" is generally used when a home exists; however, in embryogenesis, cell migration can occur in parallel with tissue formation, and in this case, this term may not be the most appropriate.
• Line 207: Since cytokines are not the only factors involved, perhaps you could mention hormones and growth factors as well.

Reviewer 2 ·

Basic reporting

Overall, the revised manuscript read a lot better than the previous version and the authors really improved the text, citations, flow, and scope of the review.

Experimental design

Aside from citing a lot of reviews, sometimes it would be better to cite the original papers that published specific discoveries. Overall, logical and more coherent flow. Some sections were repetitive (see below comment).

Validity of the findings

Good discussion on the potential use of AI as a gap/future direction. Could focus a bit more on how these assays are/aren't standardized, be more clear about are the benefits/disadvantages of using this assay, can it replace in vivo model completely?

Additional comments

Line 265 mentioned Fig 1, but it's supposed to be Fig 2.
Some acronyms not spelled out (line 268 HPC? or is it supposed to be HSPCs? Also Line 522, LT and ST.
Sometimes still unclear when using CFU-S vs CFUs. Also CFU-S is used for describing the in vivo assay, but then authors use it to refer to colony forming unit cells. Line 510 makes it sound like Baena et al discusses in vitro CFUs but that review is about in vivo CFU-S.
In "The past, current and future direction of colony-forming units assay in hematopoietic stem and progenitor cells research", a big section of it is very repetitive from introduction. Consider focusing on the in vitro assays only here and leave the history of in vivo CFU-S in the introduction when describing hematopoiesis/HSCs.
Authors refer to Table 1A-C (line 443) but I only see Table 1 with the review criteria. I do think a table with an "overview of CFUs for various adult HSPCs models derived from humans and rodent/mice", this would go well with the discussion below about different examples of research using CFUs. Same for what are supposedly Table 1B-C. Since other animal models are mentioned, similar tables for those should be included in this review.
When discussing examples of research using CFUs, please be clear what models they are using. For example, paragraph starting on Line 535, does it refer to mice or human?
Fig S3: where does this data come from? Are these clinical trials using stem cells? Do they also include more "unofficial" stem cell trials from so-called stem cell clinics?

---

## Round 0.3 · accepted · Accept

Dear Zariyantey Abd Hamid and authors, i am happy to let you know that i am now accepting your manuscript for publication. Thank you for the improvements! There are some minor typos/pleonasms that need to be fixed on the production stage. Please, proofread everything carefully and many congratulations.

Reviewer 1 ·

Basic reporting

all comments have been argued and/or taken into account by adding text. All is ok for me to publish this review.
minor modifications:
line 54: Ehrlich, 1988 . This is 1888.
Lines184-188: The 3 sentences describe the same HSC activity

Experimental design

no comment

Validity of the findings

no comment

Additional comments

thank you for taking the time to improve your review.